# Avian Antibodies as Potential Therapeutic Tools

**DOI:** 10.3390/antib14010018

**Published:** 2025-02-14

**Authors:** Mats Eriksson, Anders Larsson

**Affiliations:** 1Department of Surgical Sciences, Section of Anaesthesiology and Intensive Care, Uppsala University, SE-751 85 Uppsala, Sweden; 2NOVA Medical School, New University of Lisbon, 1099-085 Lisbon, Portugal; 3Department of Medical Sciences, Section of Clinical Chemistry, Uppsala University, SE-751 85 Uppsala, Sweden; anders.larsson@akademiska.se

**Keywords:** antibiotics, antimicrobial resistance, cystic fibrosis, egg yolk, *Gallus gallus domesticus*, hen, IgY, immunoglobulin, *Pseudomonas aeruginosa*, snake venom

## Abstract

Immunoglobulin Y (IgY) is the primary antibody found in the eggs of chicken (*Gallus domesticus*), allowing for large-scale antibody production with high titers, making them cost-effective antibody producers. IgY serves as a valuable alternative to mammalian antibodies typically used in immunodiagnostics and immunotherapy. Compared to mammalian antibodies, IgY offers several biochemical advantages, and its straightforward purification from egg yolk eliminates the need for invasive procedures like blood collection, reducing stress in animals. Due to the evolutionary differences between birds and mammals, chicken antibodies can bind to a broader range of epitopes on mammalian proteins than their mammalian counterparts. Studies have shown that chicken antibodies bind 3–5 times more effectively to rabbit IgG than swine antibodies, enhancing the signal in immunological assays. Additionally, IgY does not interact with rheumatoid factors or human anti-mouse IgG antibodies (HAMA), helping to minimize interference from these factors. IgY obtained from egg yolk of hens immunized against *Pseudomonas aeruginosa* has been used in patients suffering from cystic fibrosis and chronic pulmonary colonization with this bacterium. Furthermore, IgY has been used to counteract *streptococcus mutans* in the oral cavity and for the treatment of enteral infections in both humans and animals. However, the use of avian antibodies is limited to pulmonary, enteral, or topical application and should, due to immunogenicity, not be used for systemic administration. Thus, IgY expands the range of strategies available for combating pathogens in medicine, as a promising candidate both as an alternative to antibiotics and as a valuable tool in research and diagnostics.

## 1. Introduction

Already in 1893, Klemperer [1] showed that egg yolk from hens immunized with tetanus toxin protected mice against a lethal challenge with this toxin, thereby showing that protective neutralizing proteins were transferred into egg yolk, and thereby providing embryonic immunity. The term “IgY” was denoted in 1969 to characterize antibodies extracted from egg yolk that were different from their mammalian counterparts [2].

Immunoglobulin IgY is found in birds, reptiles, amphibians, and lungfish, and is regarded as the precursor of the mammal immunoglobulins IgG and IgE [3]. Avian antibodies are present in egg yolk, whereas mammal antibodies are derived from plasma [4]. Since there is no sanguination when IgY is collected, this procedure is compliant with both animal welfare and high values of IgY [5,6,7]. Using IgY from egg yolk has several benefits, as purification is both rapid and cost-effective, since one egg may contain approximately 80–240 IgY, and one hen may produce between 1.6 and 4.8 g of IgY monthly [8,9,10,11,12,13,14,15]. Although the therapeutic market for IgY is not identical to the one for monoclonal antibodies, the size of the global monoclonal antibody therapeutics market in terms of revenue was estimated to be worth USD 252.6 billion in 2024 and is poised to reach USD 497.5 billion by 2029, growing at a CAGR of 14.5% from 2024 to 2029 [16]. Also, minor quantities of antigen are required to obtain high titers of IgY in the yolk from immunized hens [17,18,19]. Avian antibodies are gaining increasing attention due to unique evolutionary and structural properties that may offer distinct advantages. Understanding the evolutionary divergence between avian and mammalian immune systems sheds light on the functional differences in their antibodies and the potential benefits of each for therapeutic and diagnostic purposes [20,21]. Although this review focuses on potential therapeutic effects of avian antibodies, they are so far mainly used for diagnostic purposes, both in human and animal welfare [22,23,24], as well as in environmental evaluations [25]. IgY is also frequently used in laboratorial assays utilizing avian antibodies in “particle-enhanced turbidimetric assays” (PETIA) in order to improve analytical precision, and also to reduce both the costs and the turn-around-time [21,26,27]. Furthermore, Fluorescein Isothiocyanate (FITC)-conjugated chicken antibodies are able to bind to fibrinogen, p-selectin, IgG, and von Willebrand factor, and hereby used in flow cytometry [28].

Principle actions of immunoglobulins are shown in Figure 1 and Figure 2.

### 1.1. Aims of Review

This review focuses mainly on some of the advantages of avian antibodies from therapeutic aspects. In this context, it should be noted that there is a continuously increasing problem with the number of antibiotic-resistant bacteria, which highlights the need to find alternatives to conventional antibiotics.

### 1.2. A Brief Introduction to the Avian Immune System

Essential characteristics of IgY and relevant comparisons to IgG are summarized in Table 1 and Table 2 and Figure 3 [29,30,31,32,33,34,35].

The immune system of vertebrate species mainly consists of the innate system and the adaptive (acquired) system, respectively. The innate system appears to be the evolutionarily older one. It responds rapidly through cellular mechanisms (e.g., neutrophils, macrophages, and mast cells), as well as humoral factors, such as complement. The adaptive immune system, in which antibodies play a crucial role, is more specific and establishes long-term memory but acts more slowly. Antibodies are specialized proteins produced by B cells that recognize and neutralize foreign substances, such as viruses and bacteria [36]. Mammals and birds, despite sharing a common vertebrate ancestor, have evolved distinct mechanisms for generating antibody diversity and structure. Mammalian antibodies typically belong to the immunoglobulin (Ig) G class, while birds primarily produce IgY, an antibody with structural similarities to IgG but with notable biochemical and functional differences. These differences are largely attributed to the separate phylogenetic and evolutionary pathways that mammalian and avian lineages took after diverging, probably during the late Jurassic period [37,38,39].

One major structural distinction is that avian IgY lacks the hinge region found in mammalian IgG, which affects flexibility and the binding of effector proteins (Figure 3). This structural variation does not only influences how IgY interacts with pathogens, but also reduces the likelihood of cross-reactivity with mammalian immune components, a feature that can be advantageous when developing antibodies for human therapies [40]. Avian IgY has limited complement activation and Fc receptor binding compared to mammalian IgG [41]. Avian antibodies also tend to have a broader thermal stability, potentially making them more resilient and suitable for certain therapeutic contexts [42,43]. The evolutionary pressures on avian immune systems, particularly in species with high exposure to diverse pathogens, may have driven the development of antibodies that are effective across a wider range of environmental conditions [44].

The disulfide bonds link the heavy chains covalently together. In the IgG structure, the hinge region contains multiple disulfide bonds, a characteristic feature that provides flexibility, whereas in IgY, which lacks a flexible hinge region, the disulfide bonds are positioned somewhat differently, but still serve to stabilize its configuration. Structural similarities and differences between IgY and IgG are shown in Figure 3. The variable heavy and variable light regions bind specific antigens, whereas the constant light region contributes to the antibody’s stability. C_y_1 and C_y_2 interact immunologically. C_y_3, a part of the Fc-region, is involved in the activation of complement. C_y_4 is also a part of the Fc-region, which does not only interact with cellular receptors, but also contributes to the stability of IgY [40,45,46,47].

Chicken antibodies, initially a tool to avoid false positive results by rheumatoid factor [34], turned out to counteract *Pseudomonas aeruginosa* infections in patients with cystic fibrosis [48]. In addition to structural differences, avian and mammalian immune systems generate antibody diversity through distinct processes. Hence, essentially absent cross-reactivity between avian and mammalian epitopes avoid interference of immunological techniques [49]. Mammals primarily rely on somatic hypermutation in the variable regions of antibodies to create diversity and adaptability. Birds, in contrast, utilize gene conversion mechanisms, where segments of pseudo-genes are copied into functional antibody genes, creating diversity in a way that does not involve amino acid replacements seen in mammals [50,51,52,53]. This evolutionary adaptation may contribute to the stability and consistency of avian antibodies, factors that could be beneficial in therapeutic applications where a stable and reproducible antibody response is essential [54]. Furthermore, evolutionary adaptations in avian and mammalian antibodies provide complementary tools for therapeutic applications [40,55,56].

### 1.3. Antibody Diversity

Chicken antibody diversity is created slightly differently to mammalian antibodies. It is created through a combination of mechanisms that generate variation in the immunoglobulin genes. These processes allow chicken to produce a wide array of antibodies, despite having a limited number of germline immunoglobulin (Ig) genes compared to mammals.

Chicken rely on gene conversion to create their antibody repertoire. They have a single functional variable (V) gene segment for the heavy and light chain loci. Surrounding the single functional V gene are multiple pseudogenes (non-functional gene segments) upstream of the locus. During B cell development in the bursa of Fabricius, sequences from these pseudogenes are copied into the functional V gene through gene conversion. This process introduces nucleotide diversity by replacing parts of the functional V gene with sequences from pseudogenes [57].

After gene conversion, further diversity is introduced by somatic hypermutation. This involves the introduction of point mutations in the rearranged V regions of immunoglobulin genes in activated B cells. Somatic hypermutations occur primarily after antigen exposure, and are essential for affinity maturation, where high-affinity antibodies are selected [58].

While chicken have limited germline immunoglobulin gene segments, they achieve antibody diversity primarily through gene conversion and somatic hypermutation. This is supplemented by V(D)J recombination and class switch recombination to generate antibody diversity. These mechanisms allow chicken to recognize a wide variety of antigens effectively [59,60,61,62,63].

Clonal selection of antibody formation requires a genetic process for generating antibody diversity. Two key modifications of the immunoglobulin loci facilitate the generation of antibody gene diversity; i.e., site-specific gene rearrangement and targeted deamination of deoxycytidine residues in the Ig loci, respectively [64].

## 2. Therapeutic Implications of Avian Antibodies

### 2.1. Immunotherapy

Egg yolk antibodies are good immunogens in mammals. This means that if IgY is administered repeatedly (e.g., in the lungs), there will be an immune response [65,66]. At the same time, oral administration very rarely induces an immune response in humans. Most adults eat eggs regularly without any problems or immune responses to egg yolk antibodies. Oral administration of egg yolk proteins is generally regarded as safe. Children with a known allergy against hen’s eggs or positive IgE specific to egg yolk or egg white were challenged with boiled egg yolk or egg white. Compared to egg white, egg yolk caused fewer respiratory symptoms, although gastrointestinal symptoms were more frequent [67]. However, egg yolk antibodies have obtained “Generally Recognized As Safe” status from both the US Department of Agriculture and the Food and Drug Administration [68].

Eggs have also been applied locally on the skin as a wound treatment to improve healing [69,70]. The use of egg membranes, egg white proteins, or whole eggs have been reported [71]. The risk of inducing immune response by these treatments seems to be low [72].

### 2.2. Antibacterial Treatment with IgY

A pulmonary infection with *Pseudomonas aeruginosa* is detrimental in patients with cystic fibrosis (CF), and chronic pulmonary colonization is frequent [73,74]. Hens immunized with *Pseudomonas aeruginosa* produce specific anti-*Pseudomonas* antibodies, concentrated in egg yolk [75]. In a clinical study, gargling with anti-*Pseudomonas aeruginosa* IgY prevented onset of chronic *Pseudomonas aeruginosa* lung infections in patients with CF [76]. Such antibodies appear to be safe and effective in long-term prevention of *Pseudomonas aeruginosa* infections as, even 10 years after initiated treatment, the bacteria have not turned mucoid; there is a microbial determinant of more expressed inflammation within the CF lung [77,78]. Oral passive immunotherapy with specific yolk antibodies against *Pseudomonas aeruginosa* prevents pulmonary colonization with such bacteria without any severe adverse effects [75]. Furthermore, oral prophylactic treatment with egg yolk antibodies against *Pseudomonas aeruginosa* significantly reduced the number of positive *Pseudomonas aeruginosa* in patients with CF [79]. Unfortunately, neither did bronchially instilled IgY-antibodies against *Pseudomonas aeruginosa* nor such avian antibodies, when intravenously administered in an experimental model of porcine *Pseudomonas aeruginosa* pneumonia [80,81].

Visualization of IgY production and purification is shown in Figure 4.

However, antibacterial treatment with IgY against *Streptococcus mutans* is well-established for many years. Recently, chewable tablets containing antibody IgY against *Streptococcus mutans* effectively decreased such bacterial level during orthodontic treatment [82]. Also, mouth rinse as part of passive immunization with egg yolk antibodies specific to streptococcus mutans may help to control colonization of such bacteria in the oral cavity of humans [83].

In a randomized controlled trial comprising patients with refractory *Helicobacter pylori* infections, polyclonal avian tetravalent IgY from hens immunized with *Helicobacter pylori* antigens reduced such infections [84]. The importance of such a therapeutic intervention is pointed out by Zhang and co-workers in a review [85].

#### Future Perspectives

Avian antibodies obtained from hens immunized with six different strains of *Pseudomonas aeruginosa* (PAO1, PAO3, PAO5, PAO6, PAO9, and PAO11) [75] cross-reacted between the various strains of the bacteria through binding to flagellin, being the epitope in common [86]. These findings indicate the possibility of egg yolk antibodies to have broad antibacterial implications.

In severe burns, wound infections with *Pseudomonas aeruginosa* is a frequent problem [87,88,89], especially as such colonization is associated with bloodstream infections [90]. Also, skin infections with *Pseudomonas aeruginosa* occur also in several dermatological diseases, which may demand highly advanced and specialized care, including surgical approach [91]. However, one challenge would be to administer a sufficient amount of egg yolk antibodies topically. In order to overcome this, it might be possible to establish banks of such antibodies, since it takes considerable time before immunized hens start to produce antibodies in relevant quantities [92,93,94].

In this context, it should be remembered that IgY fractions have been stored in 0.9% NaCl, 0.02% sodium azide at +4 °C for over 10 years without any significant loss of antibody titer [59]. A biobank where IgG containing anti-venom serum is stored and easily available, may serve as a model [95].

### 2.3. Antiviral Treatment with IgY

Apart from infections with bacteria, avian antibodies may also have prophylactic as well as therapeutic impact on some viral disorders in both humans and animals [94,96,97,98,99,100,101] and, more specifically, against SARS-CoV-2 [102,103,104,105,106,107,108,109,110]. Although vaccines against COVID-19 are safe and effective [111], avian antibodies, derived from egg yolk from hens immunized with the receptor-binding domain of the SARS-CoV-2 spike protein, may have additional advantages, as they are inexpensive to produce, rapidly manufactured, and easily distributed as hen-derived IgY, in contrast to vaccines, which require cold-chain storage [112]. Clinical efficacy of intranasally administered anti-SARS-CoV-2 IgY antibodies has not been proven.

A phase I study did not show any significant difference in incidence of adverse events between the investigational medical product and placebo [112]. Egg yolk IgY with neutralizing activity against pseudotyped SARS-CoV-2 exhibited, in an in vitro assay, partial competition with human angiotensin-converting enzyme 2 for the binding to S1 protein, a subunit of SARS-CoV-2 [113]. Thus, authors suggested that IgY may turn out to be of prophylactic or therapeutic tool against COVID-19 [113]. Recently, it was shown that SARS-CoV-2 neutralizing chicken egg yolk antibodies competitively block the receptor-binding domain that binds to angiotensin-converting enzyme 2, and thereby have a protective role in an animal model subjected to SARS-CoV-2 challenge [114].

Furthermore, in a mouse model, where urinary tract infection was induced by *Pseudomonas aeruginosa*, prophylactic installation of IgY derived from chicken immunized with six different *Pseudomonas aeruginosa* strains significantly reduced intravesical bacterial load [115].

### 2.4. IgY in the Treatment of Human and Animal Oro-Intestinal Infections

Egg yolk antibodies might be regarded as prophylactic immunotherapy or even adjunctive treatment in oral candida [116,117,118]. Apart from the above-mentioned studies on Helicobacter pylori, orally administered avian antibodies may also play a role in both humans and animal enteric infections as reviewed both by Carlander et al. [119] and also by Mine and co-workers [120]. From this aspect, it is crucial that IgY is resistant to the gastric barrier [121]. Avian anti-canine parvovirus 2 IgY, protected against dogs orally challenged with this virus [122].

Single-chain variable fragments IgY (IgY-scFv), a genetically engineered antibody, may produce immunoglobulins molecules with high specificity and affinity towards mammalian epitopes or antigens [123]. In an experimental study, such antibodies significantly inhibited growth of canine parvovirus [124]. A disadvantage of IgY-scFvs is their short half-life, which can be prolonged by fusing scFv with the immunoglobulin G (IgG) Fc region. Such avian IgY-scFv-mammalian IgG Fc region may be a new strategy for rapid development of antibodies in both veterinary and human medical practice [125].

### 2.5. IgY in the Fight Against Parasitic Infections

Parasites are organisms that invade a host for replication, cover and/or nutrition. Parasitic infections are more frequent in low-income countries; therefore, economically favorable options (e.g., IgY) may be a promising tool in the struggle against such infections [126]. The protozoan parasite *Trypanosoma cruzi*, causing Chagas disease. IgY against this parasite may point out the possibility of counteracting this disease [127]. In an experimental study [128] where mice were infected with *Trypanosoma cruzi*, IgY seemed to improve the immune response against this parasite. Cryptosporidiosis is an intestinal infection that causes diarrhea and, occasionally, also pneumonia in humans [129]. Highly specific avian antibodies generated against cryptosporidium parvum oocyst antigens reduced binding of this parasite to a human epithelial cell line, widely used as a model of the intestinal epithelial barrier and blocked the vivacity of *cryptosporidium parvum* [130,131].

The poultry industry has a significant economic impact in several countries. Parasitic members of the *Eimeria* family cause coccidiosis, a severe avian intestinal disease. Resistance to anticoccidials is a huge problem, although maternal immunity may compete with the best anticoccidial drugs [132], whereas IgY protected offspring chicks up to 3 weeks of age. In broiler chicks, specific IgY against multiple strains of Eimeria increased the body weight of treated animals and reduced mortality [132,133,134,135,136].

### 2.6. IgY as Treatment in Snake Bites

Each year, there are more than 100,000 casualties from snake bites due to difficulties in retrieving antivenoms. To overcome this, the WHO has established a database showing the spread of venomous snakes and their respective antivenoms. Consequently, WHO guidelines for production, control and, regulation of snake antivenom immunoglobulins have been updated [137].

Apart from conventional infections, IgY may also play a role in the struggle against snake venoms. Antivenom is the only effective treatment of certain snake venoms. Antivenoms produced by the hyperimmunization of equines may cause numerous unpredictable clinical side effects, including fatal anaphylaxis [138,139,140,141] (Figure 5); hence, there is an unmet medical need for safe and reliable antivenom treatment. Several studies have shown that IgY is efficient against various snake venoms in animal models [138,141,142,143,144,145] or patient serum [146]. These animal studies have limitations, not only due to the difference in species, but also to the fact that the amounts of both venom and antivenom should correspond to what would be reasonable to find in a snakebitten and antivenom-treated human.

### 2.7. Can Avian Antibodies Replace Antibiotics in Animal Breeding?

Overuse of antimicrobials in both humans and animals is recognized as one of the main drivers of antimicrobial resistance and an increasing prevalence of antibiotic-resistant bacteria, and it is increasingly important to find an alternative to the use of conventional antibiotics [148,149,150,151]. This is an extensive problem within the food industry, since antibiotic residues in foodstuff may have considerable effects, not only on the development of resistant bacteria, but may also have immunopathological consequences in humans [152,153].

In pigs, mortality is especially high when the suckling pig stops suckling, and immunoglobulins, previously transferred via the mother sow’s milk, must then be replaced by the piglet’s own antibody production. As the piglet’s immune system is still immature, the weaning period in native living pigs is longer than in piglet production, hereby allowing the immune system to mature before the weaning period is over. However, for cost-effective piglet production, the weaning period is shortened, and the preferred method is to strengthen the piglets through adding adequate antibodies as the weaning stops. Many classes of antibiotics used for humans are used in food animals for treatment or prevention of infection, and nearly half of all veterinarians in European countries seldom collect samples for bacterial identification and drug sensitivity tests [154,155]. Antimicrobial resistance is an extensive problem in pig farms, and it is a worrisome aspect that farm-sourced multidrug resistant *E. coli* seem to have a very high genetic propensity to spread into humans [156]. In weaning piglets challenged with E. coli K88, dietary supplementation with combinations of egg immunoglobulins and phytomolecules protected weanling piglets to almost the same extent as an antibiotic growth promoting mixture. A net effect would be a reduced antibiotic burden in such an animal setting [157].

Active immunity means stimulated production of specific antibodies towards a specific source of infection. Active immunity is created either via survival of an infectious disease or vaccination. However, since small piglets have immature immune system, their immune response may not be sufficiently effective.

In this context, it is tempting to speculate on the possibility of passive immunization through the supplementary use of a diet containing avian antibodies. Eggs can be added to the feed and given to the animal that needs an increased immunity. The advantage of the method is that the host animal—the hen—does not need to be drained of blood or slaughtered. The major advantage is that such supplementation is not an antibiotic, and that antibodies and bacteria have coexisted for millions of years without any resistance problems.

From a global aspect, it should be remembered that fish is an important source of protein, especially for poor people, and cultivation of fish is important and accounts for nearly half of the fish consumed worldwide [158]. In the aquacultural environment, an increasing number of antibiotic-resistant bacteria points out the need to find alternatives to conventional antibiotics [159,160].

White shrimp (*Litopenaeus vannamei*; Pacific white shrimp or king prawn), is commonly farmed for food, although they lack adaptive immunity, which makes them susceptible to infections. *Vibrio* spp. [161] are major causes of mortality in white shrimp. Hence, antibiotics have been commonly used in the prevention and treatment of vibriosis. Preparation and administration of egg yolk powders against *V. harveyi* and *V. parahaemolyticus* can be used for passive immunization of white shrimp protected against vibrio infections [162].

Antimicrobial resistance has emerged as a global health problem, where the lowest denominator in common is overuse and misuse of antimicrobials and especially inappropriate usage of antibiotics [163]. Passive immunization through the addition of specific antibodies against pathogenetic microorganisms may be an important tool in the struggle against antimicrobial resistance, as no such adaptation against antibodies is known to occur [164].

## 3. Concluding Remarks

Avian antibodies derived from egg yolk have broad implications within both human and veterinarian fields of medicine. There are numerous publications supporting this postulate, which may be advantageous, but also a disadvantage, as this makes it more difficult to commercialize IgY, as patenting is crucial to the pharmaceutical industry. This is a serious drawback, as IgY may replace conventional antibiotics in some circumstances and could thereby, at least to some extent, contribute to diminish the risk of increased bacterial resistance.

## Figures and Tables

**Figure 1 antibodies-14-00018-f001:**
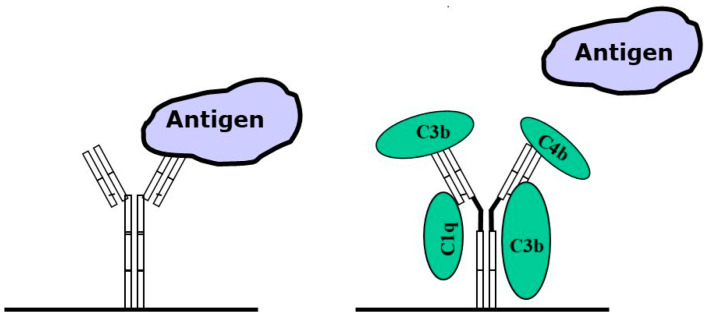
Mammalian antibodies bound to a solid phase act like immune complexes (e.g., capture antibodies in an immunological assay), and are capable of complement activation, even in the absence of an antigen. Upon activation of the complement system, coagulation components are bound to the antibodies, thus partially blocking the binding of the antigen. This occurs when analyzing fresh samples that contain an active complement system, and will lead to an underestimation of the amounts of antigen present in the sample.

**Figure 2 antibodies-14-00018-f002:**
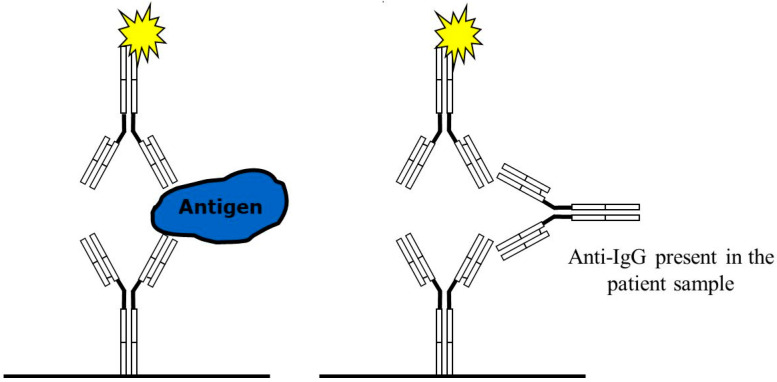
Sandwich immunoassay using chicken IgY antibodies with a specific antigen reaction (**left**) and a false positive reaction due to anti-IgG antibodies reacting with the mammalian antibody pair (**right**).

**Figure 3 antibodies-14-00018-f003:**
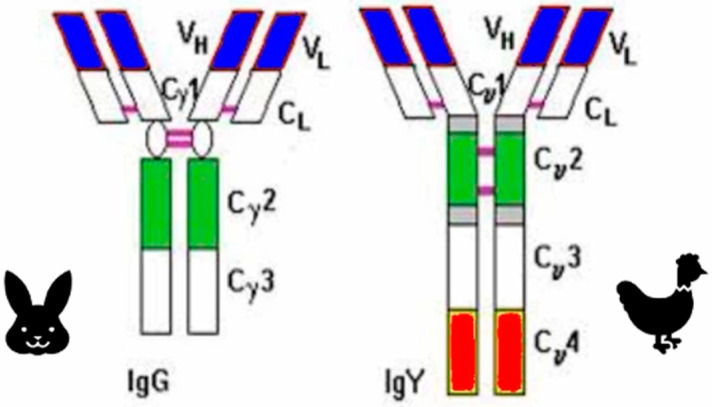
Schematic overviews of IgG and IgY, briefly showing their structural differences and similarities. The ovals at the hinge region of IgG and their corresponding parts on IgY represent disulfide bonds. Abbreviations: V_H_ (variable heavy); V_L_ (variable light); C_L_ (constant light); “C” in Cy1–4 denotes “constant”, whereas “y” (gamma) represents the heavy chain.

**Figure 4 antibodies-14-00018-f004:**
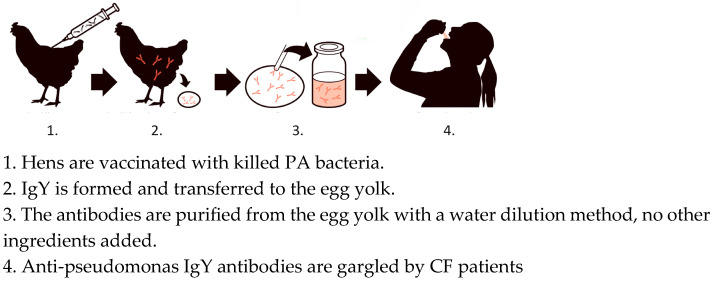
Overview of the flow pattern when anti-*Pseudomonas aeruginosa* (PA) IgY is produced and administered to patients with cystic fibrosis (CF).

**Figure 5 antibodies-14-00018-f005:**
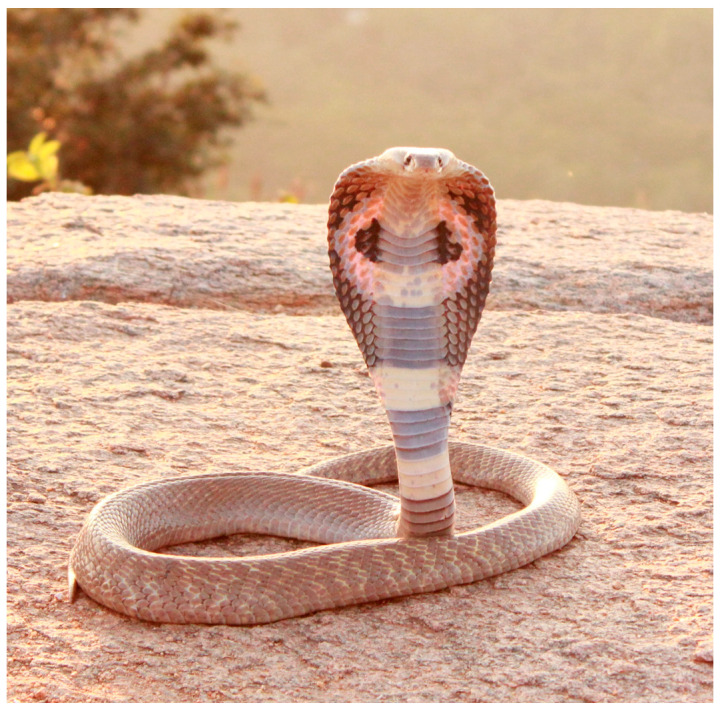
Indian Cobra (*Naja naja Atra*), mentioned in reference [138] above. Republishing is accepted when its source [147] is mentioned.

**Table 1 antibodies-14-00018-t001:** Comparison between mammalian IgG antibodies and avian IgY antibodies.

	IgG	IgY
Antibody sampling	invasive	non-invasive
Antibody yield	~100–300 mg IgG/bleeding	~80–240 mg IgY/egg
Antibody yield per month	~100–300 mg	~1.6–4.8 g
Specific antibody yield	1–10%	2–10%
Protein A/Protein G binding	Yes	No
Interference with mammalian IgG	Yes	No
Interference with rheumatoid factor	Yes	No
Binding to mammalian Fc-receptors	Yes	No
Activation of mammalian complement	Yes	No
Glycosylation	Yes	Yes, unique sequence
Molecular weight (kDA) Heavy chain Light chain	~150 ~50 ~25	~180 ~67 ~27
Isoelectric point (pH)	~7.0–9.0	~5.7–7.6

**Table 2 antibodies-14-00018-t002:** Rheumatoid factor interference. Nineteen rheumatoid-positive and ten rheumatoid-negative patients samples were added to latex particles coated with IgG from different species. All rheumatoid positive samples caused agglutination of latex particles coated with mammalian IgG, but not with particles coated with chicken IgY [28].

Antibodies	RF-Positive Samples	RF-Negative Samples
Bovine	19/19	0/10
Horse	19/19	0/10
Human	19/19	0/10
Mouse	19/19	0/10
Rabbit	19/19	0/10
Sheep	19/19	0/10
Chicken	0/19	0/10

## Data Availability

Not applicable.

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
