# Peer review of "Avian Antibodies as Potential Therapeutic Tools"

_2073-4468, 2025, doi:10.3390/antib14010018_

Round 1
Reviewer 1 Report
Comments and Suggestions for Authors
Although this review is interesting considering some points such as antibiotic resistance and the use of IgY antibodies as an alternative to mitigate resistance problems, as it is a short review, no figures or tables are shown to better summarize everything related to IgY antibodies, that is, it is important to include at least two figures and two tables.
In addition, authors should consider some points shown below:
L.31. diagnostics. "Include end point".
L.65. IgG but with notable....
L.80-81. Chicken antibodies, initially a tool to avoid false positive results by rheumatoid factor [25], turned out to counteract.
L.155. Change "since" to "for".
L.172. to administer a sufficient...
L.176-177. Check the wording of this sentence.
L.180. Change "man" to "humans".
L.231. it is increasingly important to find...
L.234. Change "man" to "humans".
L.236. In pigs, mortality....

Author Response
Reviewer 1.
Although this review is interesting considering some points such as antibiotic resistance and the use of IgY antibodies as an alternative to mitigate resistance problems, as it is a short review, no figures or tables are shown to better summarize everything related to IgY antibodies, that is, it is important to include at least two figures and two tables.
Dear Reviewer 1. Thanks for your encouraging words and willingness to improve our manuscript.
We agree. Figures and tables have been inserted into the manuscript.
In addition, authors should consider some points shown below:
L.31. diagnostics. "Include end point".
We are not quite sure about this remark, but have added the following: “Our endpoints were to establish a hierarchical evaluation of IgY within the medical field.”
L.65. IgG but with notable....
“with” is added. Thanks!
L.80-81. Chicken antibodies, initially a tool to avoid false positive results by rheumatoid factor [25], turned out to counteract.
“,” is added. Thanks!
L.155. Change "since" to "for".
“since” has been changed to “for”. We agree with this remark.
L.172. to administer a sufficient...
You are right. This is now changed.
L.176-177. Check the wording of this sentence.
This sentence has been changed and now reads: “A biobank where IgG containing anti-venom serum is stored and easily available, may serve as a model [70].”
L.180. Change "man" to "humans".
Done.
L.231. it is increasingly important to find...
The sentence has been changed according to the reviewer´s suggestion.
L.234. Change "man" to "humans".
Changed.
L.236. In pigs, mortality....
Changed. Many thanks for your remarks.

Reviewer 2 Report
Comments and Suggestions for Authors
Immunoglobulin Y (IgY), derived from hyperimmune egg yolk, is a promising passive immune agent for combating microbial infections in both humans and livestock. In this review, Mats Eriksson and Anders Larsson focus primarily on the therapeutic advantages of IgY. Overall, this review is timely and insightful. Several issues should be addressed before the manuscript can be published.
1. It is recommended to create a diagram that introduces IgY and compares it with mammalian IgG antibodies.
2. Change the title of 1.1 to “A Brief Introduction to the Avian Immune System”.
3. The application of IgY in the fight against parasitic infections also needs to be discussed and introduced.
4. Although the author's focus is on summarizing the potential therapeutic value of IgY, the applications of IgY in diagnosis should not be overlooked and need to be briefly introduced.
Author Response
Reviewer 2
Immunoglobulin Y (IgY), derived from hyperimmune egg yolk, is a promising passive immune agent for combating microbial infections in both humans and livestock. In this review, Mats Eriksson and Anders Larsson focus primarily on the therapeutic advantages of IgY. Overall, this review is timely and insightful. Several issues should be addressed before the manuscript can be published.
Dear Reviewer 2. Many thanks for kind wordings, interest in this review and your willingness to help us to improve our manuscript.
- It is recommended to create a diagram that introduces IgY and compares it with mammalian IgG antibodies.
We realize that the lack of figures/tables is a shortcoming in this review. Figures with IgY and IgG as well as a table describing main differences between the two types of antibodies have been inserted.
- Change the title of 1.1 to “A Brief Introduction to the Avian Immune System”.
This heading describes this section and the title is changed, even if this section now is 1.2.
- The application of IgY in the fight against parasitic infections also needs to be discussed and introduced.
A new 2.5 section has been added.
- Although the author's focus is on summarizing the potential therapeutic value of IgY, the applications of IgY in diagnosis should not be overlooked and need to be briefly introduced.
In the “1. INTRODUCTION” section, the diagnostic use of IgY is now extended and new references are inserted.

Reviewer 3 Report
Comments and Suggestions for Authors
Dear authors
I hope you are all doing well. Regarding the revision of manuscript number antibodies-3396641 entitled "Avian antibodies as therapeutic tools - A comprehensive review". It really is an interesting review. However, some comments should be answered for better improvement.
1- Pseudomonas aeruginosa in lines 139-177 and196 should be in italics.
2- Streptococcus mutans in lines 154 and 156 should be italicized.
3- Helicobacter pylori on line 201 should be italicized.
4- V. harveyi and V. parahaemolyticus on line 272 should be italicized.
5- One or two innovative figures are essential to break the reader's boredom.
6- A comparison table between human IgG and IgY is needed.
Author Response
Reviewer 3
Dear authors
I hope you are all doing well. Regarding the revision of manuscript number antibodies-3396641 entitled "Avian antibodies as therapeutic tools - A comprehensive review". It really is an interesting review. However, some comments should be answered for better improvement.
Dear Reviewer 3, many thanks for your kind words and your valuable input.
2- Streptococcus mutans in lines 154 and 156 should be italicized.
3- Helicobacter pylori on line 201 should be italicized.
4- V. harveyi and V. parahaemolyticus on line 272 should be italicized.
You are right about these remarks and these names, as well as some recently added microorganisms, are now in italics.
5- One or two innovative figures are essential to break the reader's boredom.
This has been accomplished.
6- A comparison table between human IgG and IgY is needed.
We agree. We have inserted both figures and tables on this topic.

Reviewer 4 Report
Comments and Suggestions for Authors
- Structure and Organization:
a) Fragmented Future Perspectives sections appear in multiple places:
- Lines 162-177 (under antibacterial treatment)
- Lines 470-471 (near conclusion) These should be consolidated into one comprehensive section.
b) Flow issues:
- Lines 117-121 ("Aim of review") comes after substantial content and should be moved earlier
- Lines 199-279 discuss multiple topics without clear transitions between IgY treatment topics
- Scientific Content:
a) Antibody diversity section needs expansion:
- Lines 95-116 could include recent findings about:
- Gene conversion mechanisms
- Somatic hypermutation
- Current understanding of selection processes
b) COVID-19 discussion limitations:
- Lines 181-196 discuss SARS-CoV-2 applications but don't adequately address:
- Storage stability challenges
- Production scale-up issues
- Comparative efficacy data
- Technical Details:
a) Statements needing updated citations:
- Lines 229-231: "Overuse of antimicrobials in both humans and animals is recognized as one of the main drivers of antimicrobial resistance" needs recent supporting data
- Lines 232-235: Claims about antibiotic residues need current references
b) Missing quantitative data:
- Lines 46-48: "purification is both rapid and cost-effective, since one egg may contain more than 100 mg IgY" should include comparative cost analysis
- Lines 174-176: Storage claims ("stored for up to 10 years") need supporting data
- Language and Clarity:
a) Overly complex sentences:
- Lines 63-68: "Mammals and birds, despite sharing a common vertebrate ancestor, have evolved distinct mechanisms for generating antibody diversity and structure..." Could be split into two sentences.
b) Terminology inconsistencies:
- "Avian antibodies" (Line 49) vs "chicken antibodies" (Line 80)
- "IgY" vs "egg yolk antibodies" used interchangeably
- Visual Elements Needed:
a) Missing crucial figures:
- No structural comparison between IgY and IgG despite discussion in lines 69-74
- No visualization of production process described in lines 46-48
- Methods Section:
Add after Line 35:
- Literature search methodology
- Inclusion/exclusion criteria
- Databases consulted
- Time period covered
- Specific Section Improvements:
Abstract (Lines 12-31):
- Add specific success rates/efficacy data
- Include brief mention of production costs
- Note regulatory status
Introduction (Lines 35-45):
- Add clear objectives statement
- Provide context for therapeutic antibody market
Results Organization:
- Consolidate therapeutic applications (Lines 138-198)
- Group similar applications together
- Add comparative efficacy data where available
- Reference Updates Needed:
Examples of older references needing updates:
- Line 299: Klemperer 1893 should be supplemented with modern interpretations
- Lines 314-315: Bhanushali 1994 could be updated with newer purification methods
Author Response
Dear Reviewer 4,
Many thanks for your review and your constructive criticism, which has helped us to further improve this review. We hope that you will accept our answers and explanations. In a few cases we have not been able to change the manuscript according to your suggestions, but we have always responded to the best of our ability.
- Structure and Organization:
- a) Fragmented Future Perspectives sections appear in multiple places:
- Lines 162-177 (under antibacterial treatment)
- Lines 470-471 (near conclusion) These should be consolidated into one comprehensive section.
Our focus in the “Future perspectives” section was on anti-bacterial treatment, but since anti-venom serum was mentioned, we find it reasonable to add the reference “Wen et al, 2012” into this section as well.
- b) Flow issues:
- Lines 117-121 ("Aim of review") comes after substantial content and should be moved earlier
“Aims of review “has been moved to the final part of the INTRODUCTION section and is now denoted 1.1
- Lines 199-279 discuss multiple topics without clear transitions between IgY treatment topics
In: “Aims of review”, we stated the following:
This review focuses mainly on some of the advantages of avian antibodies from therapeutic aspects. In this context it should be noted that there is a continuously increasing problem with the number of antibiotic-resistant bacteria, which highlights the need to find alternatives to conventional antibiotics.
We think that sections 2.4., 2.5., and 2.6. focus on these topics.
In section 2.4. both human and canine infections are discussed, since IgY may also play a role in counteracting both humans and animal enteric infections.
2.5. Snake bites should be a clear transition.
Section 2.6 describes the possible use of avian antibodies in animal breeding, in order to replace excessive use of antibiotics. Here we discuss the impact of antibiotic use on the environment and its economic implications. This part could be extended to a review of its own, but we think that the text and the references should be a sufficient illustration of these problems.
However, as the Reviewer remarks transitions could be improved. We hope that these changes improve the text flow.
- Scientific Content:
- a) Antibody diversity section needs expansion:
- Lines 95-116 could include recent findings about:
- Gene conversion mechanisms
- Somatic hypermutation
- Current understanding of selection processes
Although the focus of our review is on therapeutic effects of IgY, a brief notice on these issues could be interesting. Therefore, a separate para and a reference have been inserted.
- b) COVID-19 discussion limitations:
- Lines 181-196 discuss SARS-CoV-2 applications but don't adequately address:
- Storage stability challenges
- Production scale-up issues
- Comparative efficacy data
We presume that these remarks refer to IgY in its prophylactic/therapeutic use against SARS-CoV-2.
We are sorry, but we have not found any such information of IgY in this context. If the reviewer is aware of any such publications, we will be happy to add them to our review.
Storage stability of therapeutic antibodies is a complex issue as it not only involves the stability of the antibodies but also the risk of microbial contamination. The contamination risk varies depending on the administration method. Intravenous administration will have a much higher requirement than an oral administration. Please, see also below.
Production scale-up issues very much depends on the purity of the antibody production. If we take the example Intralipid which is an egg yolk based intravenous lipid formulation for intravenous use the production utilizes approximately 600 000 eggs per day. A large US poultry unit has approximately 1 million hens producing more than 600 000 eggs per day. The Swedish company Källbergs in Töreboda ( https://www.kallbergs.se), handles more than 1 million eggs per day (cracking, separating and spay drying) and there are much larger units in other countries. The pseudomonas treatment study utilized half an egg per daily patient dose. Thus, it is no problem reaching production volumes in the excess of 1 million doses per day.
However, as the Reviewer remarked (below), we should provide context for therapeutic antibody market. This is now briefly mentioned in the Introduction section and a reference s inserted.
- Technical Details:
- a) Statements needing updated citations:
- Lines 229-231: "Overuse of antimicrobials in both humans and animals is recognized as one of the main drivers of antimicrobial resistance" needs recent supporting data
- Lines 232-235: Claims about antibiotic residues need current references
Since understanding of overuse of antimicrobials is crucial, we have added three references supporting the postulate that IgY may contribute to reduction of unnecessary use of such drugs.
- b) Missing quantitative data:
- Lines 46-48: "purification is both rapid and cost-effective, since one egg may contain more than 100 mg IgY" should include comparative cost analysis
A reference on cost-effectiveness in the production of IgY is inserted.
- Lines 174-176: Storage claims ("stored for up to 10 years") need supporting data
The sentence has been rephrased and a modified quote from the article supporting this statement as been inserted.
There are no other such publications as this is not in the line of interest of the manufacturers. AL is still using the antibodies he produced in 1983 (40 years ago) but we do not want to cite that as we are using the antibodies but we do not have scientific data on the exact stability apart from that they are still active in e.g. ELISA. The antibodies have been stored at +4C as purified antibodies in phosphate buffer with azide. Please, see also our comments above on storage stability challenges.
- Language and Clarity:
- a) Overly complex sentences:
- Lines 63-68: "Mammals and birds, despite sharing a common vertebrate ancestor, have evolved distinct mechanisms for generating antibody diversity and structure..." Could be split into two sentences.
Lines 62-68 read: “Mammals and birds, despite sharing a common vertebrate ancestor, have evolved distinct mechanisms for generating antibody diversity and structure. Mammalian antibodies typically belong to the immunoglobulin (Ig) G class, while birds primarily produce IgY, an antibody with structural similarities to IgG but notable biochemical and functional differences. These differences are largely attributed to the separate phylogenetic and evolutionary pathways that mammalian and avian lineages took after diverging, probably during the late Jurassic period [17-19].”
We do not understand which sentence the reviewer want us to split into two. Guidance would be appreciated.
- b) Terminology inconsistencies:
- "Avian antibodies" (Line 49) vs "chicken antibodies" (Line 80)
In line 49, we used the term “Avian antibodies” to point out the evolutionary aspect, which does not only refer to chickens, but to birds from a broader aspect.
In line 80, we wrote “chicken antibodies”, since the reference specifically refers to chicken antibodies.
- "IgY" vs "egg yolk antibodies" used interchangeably
This is now more uniform, except when we wanted to differ or specify the type of antibody.
- Visual Elements Needed:
- a) Missing crucial figures:
- No structural comparison between IgY and IgG despite discussion in lines 69-74
We agree with the Reviewer. The schematic structures of both IgY and IgG have been inserted.
- No visualization of production process described in lines 46-48
We agree with the Reviewer that such a visualization could be helpful for understanding. Since the main therapeutic implication of IgY has been in cystic fibrosis and pulmonary infection with Pseudomonas aeruginosa, the picture was inserted in this context.
Methods Section:
Add after Line 35:
- Literature search methodology
Since we (especially A.L.) have presented science on IgY, background knowledge was at hand.
Please, see also below.
- Inclusion/exclusion criteria
No specific terms apart from what we considered as relevant were used.
- Databases consulted
Apart from our own knowledge, Pubmed was the essential source, supplemented with Google when necessary.
- Time period covered
Our focus was to find relevant references.
- Specific Section Improvements:
Abstract (Lines 12-31):
- Add specific success rates/efficacy data
- Include brief mention of production costs
- Note regulatory status
?
There is no room for success rates/efficacy data as this would be based on individual studies.
Production cost? The egg production cost is less than 0.2 USD. If it is a pharma grade product the price will be governed by the clinical value and could increase thousandfold compared to the yolk price. The production cost is also depending on the pharma requirements and are therefore impossible to give a general mention. As far as we know there are no registered egg based formulations.
Introduction (Lines 35-45):
- Add clear objectives statement
Section “1.1 Aims of review” is inserted.
- Provide context for therapeutic antibody market
Information on this is now inserted in the Introduction section (please, see also above 2b)
The size of global monoclonal antibody therapeutics market in terms of revenue was estimated to be worth $252.6 billion in 2024 and is poised to reach $497.5 billion by 2029, growing at a CAGR of 14.5% from 2024 to 2029.
Results Organization:
- Consolidate therapeutic applications (Lines 138-198)
This part has been restructured and a section: 2.5. IgY in the fight against parasitic infections” is inserted.
- Group similar applications together
- Add comparative efficacy data where available
No randomized clinical trial on the efficacy of IgY are, to the best of our knowledge, published.
- Reference Updates Needed:
Examples of older references needing updates:
- Line 299: Klemperer 1893 should be supplemented with modern interpretations
The first reference (Klemperer, 1893) was cited merely to put the use of egg yolk derived from immunized hens in a historical perspective. We believe that the manuscript contains several modern interpretations of the protective effect of what we now call, IgY. However, if the reviewer demands that this reference should be omitted, we are, of course, willing to do so.
In order to avoid misunderstandings, the first sentence in the Introduction section is now rephrased.
- Lines 314-315: Bhanushali 1994 could be updated with newer purification methods
We agree with the reviewer that there is a need for more updated references on the purification of antibodies. Consequently, the references are updated.

Round 2
Reviewer 1 Report
Comments and Suggestions for Authors
The authors considered each of the suggestions made in the first review, but new recommendations are shown below:
1) Table 2 may not be necessary, it doesn't really show much.
2) Review the format of Table 3 as its title seems to be part of the table content.
3) Figure 3 should have a single title and not as shown in the review "Figure 3a. Schematic structure of IgY, and Figure 3b. Schematic structure of IgG.
4) In order to have the same font, it is suggested to change the font format to the same font as the text in Figure 4.
5) Figure 5 is named as Figure 3, which is incorrect (L.305) and is also not referenced in the text.
L.72. Change "is" to "are"
L.180. Change "has" to "have"
Author Response
Reviewer 1 Round 2
The authors considered each of the suggestions made in the first review, but new recommendations are shown below:
Dear Reviewer 1, we are pleased to note that you found our answers in the first round to be adequate.
- Table 2 may not be necessary, it doesn't really show much.
We agree that the information in Table 2 already has been declared in the text.
Although we think that it might be convinient to summarize information in tables, this one has been omitted.
- Review the format of Table 3 as its title seems to be part of the table content.
In order to increase the clarity of this table, the line breaking is removed.
- Figure 3 should have a single title and not as shown in the review "Figure 3a. Schematic structure of IgY, and Figure 3b. Schematic structure of IgG.
The legend presenting Figure 3 is now rephrased to a single sentence.
- In order to have the same font, it is suggested to change the font format to the same font as the text in Figure 4.
Figure 4 is now in Palatino Linotype.
5) Figure 5 is named as Figure 3, which is incorrect (L.305) and is also not referenced in the text.
This figure is now correctly denoted Figure 5. The figure and a sentence are now inserted in the text flow.
In the legend to Figure 5 a relevant reference, mentioned in the text, is given. The legend is slightly rephrased to increase clarity.
L.72. Change "is" to "are"
L.180. Change "has" to "have"
Changes are performed.

Reviewer 2 Report
Comments and Suggestions for Authors
Following the previous round of reviews and the author's revisions, the quality of the manuscript has significantly improved.
Author Response
Reviewer 2 Report 2
Following the previous round of reviews and the author's revisions, the quality of the manuscript has significantly improved.
Dear Reviewer 2. Many thanks for your valuable input and your kind words.
